# Concrete–Representational–Abstract (CRA) Instructional Approach in an Algebra I Inclusion Class: Knowledge Retention Versus Students' Perception

**Sherri K. Prosser** [1,*] and **Stephen F. Bismarck** [2]

1 Department of Educational Specialties, Austin Peay State University, Clarksville, TN 37044, USA
2 Department of Education, University of South Carolina Upstate, Spartanburg, SC 29303, USA; sbismar@uscupstate.edu
* Correspondence: prossers@apsu.edu; Tel.: +1-386-314-3015

**Abstract:** Mathematical manipulatives and the concrete–representational–abstract (CRA) instructional approach are common in elementary classrooms, but their use declines significantly by high school. This paper describes a mixed methods study focused on knowledge retention and perceptions of students in a high school Algebra I inclusion class after a lesson on square roots using a novel algebra manipulative. Twenty-five students in a high school Algebra I inclusion class engaged in an interactive lesson on square roots paired with the manipulative to support their conceptual understanding. Participants completed a pretest, a post-treatment questionnaire, and a delayed post-test. The two-sample *t* test showed a significant difference in students' pretest–post-test scores. However, conventional content analysis of the questionnaires showed that most students did not believe the CRA instructional approach supported their learning. Implications include increased use of manipulatives to teach abstract algebraic topics to support students' conceptual understanding and destigmatizing the use of manipulatives in secondary mathematics classrooms.

**Keywords:** algebra manipulative; inclusion class; concrete–representational–abstract approach; high school algebra; secondary mathematics; student perceptions





## 1. Introduction

"Our goal must be to develop the talents of all to their fullest. Attaining that goal requires that we expect and assist all students to work to the limits of their capabilities" [1] (p. 12).

The quotation above is taken from the seminal report *A Nation at Risk* and still applies regarding the need for educators to implement instructional practices that assist all students in reaching their fullest potential [1]. As of 2017–2018, the most recent data available, approximately 14% of all public school children in the United States are educated under the Individuals with Disabilities Act [2]. Students with a specific learning disability (i.e., learning disability or LD) constituted the largest percentage of students with disabilities, at 33.6% [2]. The call for equitable treatment of and access for all students has echoed throughout the years in mathematics education reform [3–5].

When examining high school mathematics classrooms that include students with LDs, mathematics teachers focus on providing accommodations to students with LDs that range from additional time to complete assignments to alternative homework assignments and to the use of calculators [6]. These accommodations can be attributed to interventions that target procedural mimicry and recall of facts [7]. A growing body of research studies examining the effectiveness of using hands-on and interactive materials with high school students with LDs provide consistent support for instructional practices that go beyond the accommodations and interventions referenced above [8,9]. Within the mathematics

education community, hands-on and interactive materials often include what are referred to as "manipulatives".

Manipulatives are objects (either virtual or concrete) used to represent abstract mathematical ideas concretely [10]. The conceptual grounding for using manipulatives originates from aspects of constructivist theory that connect students' concrete perceptions and experiences of the world and abstract thinking [11]. The constructivist perspective provides a promising approach as students with LDs have difficulty generalizing learned material and conceptualizing abstract algebraic concepts and tasks [7,12,13]. Manipulatives are concrete objects that students can arrange, partition, and group in ways that assist them in abstract thinking associated with specific mathematical concepts [14].

With the increased availability of electronic devices (e.g., computers, tablets, and interactive whiteboards) in classrooms, the use of virtual manipulatives has also increased. Virtual manipulatives are digital representations of concrete objects. Studies have compared the effectiveness of concrete manipulatives to virtual manipulatives and found them to be equally effective [15–17]. Specifically, Westenskow and Moyer-Packenham [17] examined the use of both concrete and virtual manipulatives for students with mathematical learning disabilities and found that both types of manipulatives provided evidence of statistically significant knowledge gain on a variety of fraction concepts; concrete manipulatives were favored approximately half of the time, and virtual manipulatives favored the other half.

However, it should be noted that using manipulatives does not elicit the automatic learning of mathematical concepts. Ball's [18] article on using mathematical manipulatives with elementary school students puts this notion of mathematical understanding in perspective: "[U]nderstanding does not travel through the fingertips and up the arm. Although concrete materials can offer students context and tools for making sense of the content, mathematical ideas really do not reside in cardboard and plastic materials" (p. 47). Manipulatives need to be implemented in the classroom using appropriate instructional practices. After a 72 h rigorous professional development program specific to a constructivist approach that included manipulatives, for example, teachers of mathematics expressed a belief that students learn abstract topics best when engaged with hands-on activities but noted classroom management (e.g., distractions due to manipulatives) as a barrier to implementation of this approach [19].

One instructional approach that uses concrete models, such as manipulatives, is the concrete–representational–abstract (CRA) approach, which is sometimes referred to as the graduated instructional sequence or the concrete–pictorial–abstract approach. The three parts of the CRA instructional approach build upon each other. The CRA sequence begins with students using the manipulative as they work on a task (i.e., the concrete stage). Once students master the concrete stage, they can create a pictorial display of a completed task with the manipulative (i.e., the representational stage). In the last stage (i.e., the abstract stage), students use numerical or algebraic symbols to facilitate abstract reasoning [20]. For example, students could use base-ten blocks to display a multidigit multiplication problem at the concrete stage, a drawing of the base-ten blocks to comprise the representational stage, and Arabic numerals and symbols to show the same problem during the abstract stage (see Figure 1).

In the early 1980s, Singapore's Ministry of Education began advocating for the implementation of the CRA instructional approach [21]. The strategy has become a staple of teaching mathematics in Singapore and is the grounding behind the country's success on several international mathematics achievement assessments [22]. The CRA sequence has been used while teaching with manipulatives for decades in the United States and has been shown to be an effective approach in mathematics classrooms that include students with LDs [9,23]. Unfortunately, using manipulatives and, therefore, the CRA approach has not been implemented regularly in high school mathematics classrooms [24], likely due to the misconception of the effectiveness of manipulatives with older students [25] and that manipulatives use is distracting [19]. Implementing instructional practices that assist all students includes examining manipulatives that target high school concepts through

concrete and visual representations using the CRA approach. Therefore, this article will examine the implementation of the CRA approach with an Algebra I inclusion class, focusing on students' knowledge retention and perception of their learning.

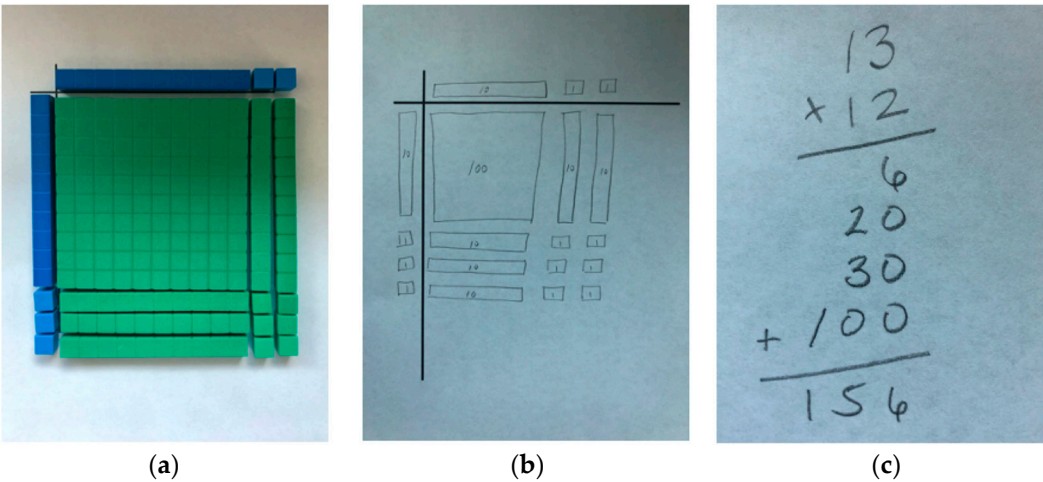

**Figure 1.** A multidigit multiplication problem, 12 × 13, is displayed using each stage of the CRA approach: (**a**) Panel 1 shows base-ten blocks, which is the concrete stage; (**b**) Panel 2 shows a diagram, which is the representational stage; and (**c**) Panel 3 shows symbols that model the problem, which is the abstract stage.

## 2. Perspectives on Manipulatives Use and Need for the Study

A common view among researchers is that instructional practices involving manipulatives are beneficial for young students but unnecessary for older students [25,26]. Using manipulatives in the mathematics classroom has typically been associated with a child's stage of cognitive development [27]. Cognitive development theories suggest that there are developmental stages in children's ability to think and learn as they age [28–30]. For this paper, we focus on the progression from concrete to formal or abstract operations [30,31]. In this progression, children go from using concrete materials to assist them in learning mathematics to the ability to more fully rationalize abstraction while learning mathematics [32]. As a result of this developmental-based progression, hands-on materials and manipulatives are predominantly used to help students learn mathematics in elementary grades [25,26].

Results from a survey conducted by Swan and Marshall [24] reported that using such hands-on materials is almost entirely abandoned by the time students reach high school. Even in middle schools, for example, only 17% of mathematics teachers use manipulatives *frequently* or *very frequently* [33]. It is important to note that cognitive development theory suggests that adolescents develop the ability to more fully rationalize abstraction, but not that concrete materials hinder them in the development of knowledge [34]. The overgeneralization of this theory has contributed to a view of children's cognitive development as inflexible and dichotomous [34].

Research suggests that nearly two-thirds of 17-year-olds (i.e., high school juniors and seniors) have yet to move past the concrete level of thinking and will struggle to formulate abstract thought patterns using purely symbolic representations and making generalizations with little context [32,35]. Furthermore, national groups involved in mathematics education advocate for all students to be actively engaged in the learning of mathematics at all grade levels [4,5,36]: "Students at all grade levels can benefit from the use of physical and virtual manipulative materials to provide visual models of a range of mathematical ideas" [5] (p. 82).

Limited research exists on implementing the CRA approach in high school mathematics courses (e.g., algebra, geometry, trigonometry, and calculus). Bouck and Park [37] conducted a systematic literature review of studies between 1975 and 2017 on using manipulatives to support students with LDs; 29 articles explored using manipulatives through the CRA

approach. Of those 29 articles, 5 were identified as targeting secondary mathematics students [13,38–41], 4 of which found that all students increased their mathematical knowledge, skills, or both. Maccini and Hughes [39] found that five of the six student participants improved on all mathematical tasks. This small set of studies presents a promising approach to teaching mathematics to high school students with LDs and requires further investigation.

The low number of studies on high school students' manipulatives use may also be a function of the small number of manipulatives available that target abstract concepts typically taught in the high school mathematics curriculum. Linking concrete materials to abstract representations has presented a significant challenge for educational research [42]. Inherent in this challenge is the dual-representation hypothesis. The dual-representation hypothesis occurs when "symbols are simultaneously objects in their own right and representations of something else" [42] (p. 156). For example, when using algebra tiles, the variable $x$ is represented by the length of a rectangle. The fact that the rectangle has a fixed length may make it harder for the student to focus on the representation of the length, as $x$ is typically presented as an unknown value. This dual representation presents a need to examine and evaluate different manipulatives that target high school concepts through concrete and visual representations while minimizing unnecessary complexities and dual representations.

Indeed, one response-to-intervention recommendation for elementary and middle school students is using concrete manipulatives when visual representations are insufficient for student understanding of the abstract [43]. In fact, the systematic use of manipulatives and visual representations in 13 randomized controlled trials showed moderate evidence for improving students' conceptual understanding [43]. However, others have argued for the appropriateness of manipulative use when introducing new mathematics topics to enhance conceptual understanding and problem solving [44–46]. After an updated review of mathematics interventions for secondary students with learning disabilities, Maccini et al. [46] called for future studies to be conducted within general education classrooms to enhance generalizability and to address middle or high school mathematics topics instead of remedial topics. The focus of this mixed methods study is to determine the knowledge acquisition and retention of knowledge by students after a CRA lesson about an abstract algebra concept. The following research questions guided this study:

1. To what extent will students in a high school Algebra I inclusion class retain the knowledge of simplifying square roots after being instructed using the CRA instructional approach?
2. How will students in a high school Algebra I inclusion class describe the effectiveness of using a mathematical manipulative to learn about a specific mathematical procedure?

## 3. Materials and Methods

The data highlighted in this article were from a larger study involving 4 teachers and 212 students within 10 college preparation mathematics classes (five Algebra I and five Geometry) at a suburban high school in the southeastern United States. The larger study was a pretest-delayed post-test control group experimental design with three randomly selected algebra classes and three randomly selected geometry classes receiving instruction using the CRA approach (treatment). The remaining two algebra classes and two geometry classes received traditional, didactic instruction (control). Each student in the treatment group received an anonymous open-response questionnaire to complete after the treatment lesson. Approximately one month after the lesson, the post-test was administered to students in both the control and treatment groups.

During the analysis of the pretest–post-test data, the second author discovered that one of the Algebra I classes from the treatment group showed a statistically significant difference that was much greater than any of the other classes. For example, the analysis of two Algebra I treatment classes produced $p$ values of 0.017 and 0.0031, but the other class had a $p$ value less than 0.0001. The researcher contacted the teacher of record to discuss the outlier findings and was informed that the class was an inclusion class. Because this information was not known prior to the study, there was no control group specific to the

inclusion class. The statistically significant results in the larger study led to this deeper investigation into the qualitative responses of the students in the inclusion class. This type of retrospective analysis has been referred to as "unmotivated looking"—a term coined by Sacks [47] to describe qualitative analysis when the results are expected to have practical application [48]. For example, educational researchers have applied unmotivated looking to (re)analyze transcripts after unexpected but salient findings in their initial analysis [49].

This study applied a complementarity mixed methods design, which "seeks elaboration, enhancement, illustration, clarification of the results from one method with the results from the other method" [50] (p. 259). Unlike the more commonly used triangulation designs, in which two methods assess the same aspect of a phenomenon, qualitative and quantitative methods in a complementarity design assess different but overlapping facets of a phenomenon [50].

### 3.1. Context and Participants

The mid-sized high school enrolled 911 students; 53% were deemed proficient in mathematics as measured by state standardized test scores. The school receives Title I funding, with 50% of the population considered economically marginalized [51]. The student racial/ethnic identification is 78% White, 12% African American, 5% Hispanic, and 5% Asian or American Indian.

College preparation and honors were the only two levels of Algebra I classes offered at this school. The class of interest for this article, the Algebra I inclusion class, comprised 25 students, of which at least 40% were identified as having a high-incidence disability (e.g., specific learning disability, speech impairment, language impairment, or other health impairment). The teacher of record was not at liberty to disclose students' specific diagnoses.

### 3.2. Lesson

During the lesson, the second author facilitated the entire lesson and took field notes while students worked in small groups. The teacher of record was present in the classroom for the entirety of the lesson. The teacher sat in the back of the classroom, away from the students, and observed the lesson.

The students were taught one lesson on simplifying square roots by the researcher (not the teacher of record) over one 90 min class block using the CRA instructional approach. The lesson began with a 10 min discussion about square roots and their connection to side lengths of squares and ended with introducing the students to the geometric definition of a square root, the length of the side of a square with a given area. Students were placed in pairs, pre-assigned by the teacher of record, and provided with instructions regarding the manipulative and the activity sheet. Students were then instructed to begin working on the first task. The researcher acted as a facilitator, monitoring and answering technical questions regarding the manipulative. After each task, the researcher asked groups to present their solutions and provided time for whole group discussion and connections before moving on to the next task. The first three tasks focused on developing conceptual understanding through the use of the manipulative (i.e., concrete stage), the next two tasks transitioned to drawing representations of the manipulative (i.e., representational stage), and the remaining three tasks provided opportunities to connect the concrete and representational stages to the numerical representation (i.e., abstract stage). This was performed by completing a table, making conjectures, and analyzing the numerical progression, which is traditionally associated with simplifying square roots. Neither group of students received additional instruction on the concept.

#### The Manipulative

As both concrete and virtual manipulatives have been found to be equally effective [17], the second author in this study used a concrete version of the manipulative; a virtual version is being developed. The manipulative, which was piloted and refined prior to the larger study [52], has since been cited in multiple state standards documents [53–55] and has been used

in mathematics professional development grant projects [56]. To initiate the CRA approach, the second author created concrete (i.e., tactile) square manipulatives by printing, laminating, and cutting out squares with whole number areas ranging from 2 cm$^2$ through 10 cm$^2$.

Students can select square tiles of a particular size and arrange them in an array to create squares of larger areas, such as those shown in Figure 2. This process allows students to physically model finding perfect square factors rather than watching the teacher model the process. The second author created an activity sheet that provided students with different-sized squares to partition using the manipulative. This activity sheet provided students with opportunities to think and write about the partitions, generalize, and connect the visual representation to the corresponding numerical process.

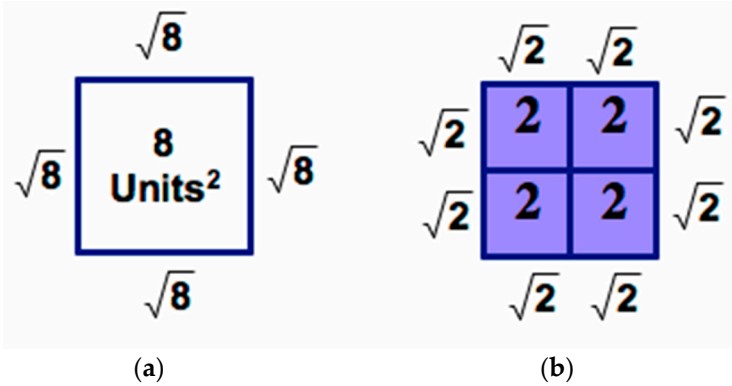

(a)  (b)

**Figure 2.** These diagrams are representations that show the equivalence of (**a**) $\sqrt{8}$ to (**b**) $2\sqrt{2}$.

The manipulative minimizes dual representation because the representation of the manipulatives (squares) is directly related to the task (simplifying square roots). The manipulatives are represented as squares of specific whole number areas (in cm). It is critical that all of the squares represent accurate measurements for this model to work. To complete the task, students do not need to interpret these squares in any other context, size, or representation. In fact, it is essential that students see the manipulatives as squares with given areas that correspond to their side lengths. The properties of the square manipulatives are important to their use and, therefore, do not present dual representation. Minimizing dual representation with this type of design has been hypothesized to facilitate transfer and retention [57].

### 3.3. Instrumentation

Two paper-and-pencil instruments were used for this study: a pretest–post-test and an anonymous open-response questionnaire. The pretest–post-test consisted of three procedural items and two relational items [58]. The procedural items asked students to simplify a given square root. The relational items focused on student explanation, the connection between perfect square factors, and the simplification process. The first relational item asked students to explain why a given square root could not be simplified. The second relational item asked students to identify an error in the simplification process, explain the error, and correct the error. The total score of the assessment was five points (one point for each item). All the questions targeted knowledge needed when simplifying square roots and not knowledge specific to the manipulative or activity.

The questionnaire consisted of two items, each followed by several lines for students to write their responses. The first question asked the students how the activity was helpful to them and to provide support and explanation for their response. The second question asked students whether they thought the activity would help them remember how to simplify square roots and why.

*3.4. Procedure*

3.4.1. Data Collection

Data were collected using the pretest, questionnaire, and delayed post-test. The five-question pretest was administered by the teacher of record the day before the intervention lesson, and the post-test was administered approximately one month after the lesson had been taught. This length of time between the lesson and the post-test was chosen to measure knowledge retention [59]. All 25 students completed both the pretest and post-test. The open-response questionnaire was provided to the students after the lesson by the teacher of record. Students were given approximately 10 min to complete the questionnaire, and all 25 students responded to each question.

3.4.2. Quantitative Data Analysis

Pretest–post-test scores for each treatment group were analyzed to determine if students ($N = 25$) showed a significant score increase using matched pair *t* tests. Normality was confirmed for the group using histograms and *Q-Q* plots in the computer program SAS. The *t* test examined if there was a significant gain from the pretest to the post-test, making the test one-sided. The alpha value used to determine significance was 0.01.

3.4.3. Qualitative Data Analysis

The open-response questionnaire was analyzed using conventional content analysis [60]. Conventional content analysis is often used when the goal of a study is to describe a phenomenon where limited information is available related to the phenomenon under study, and open-ended responses are available to form a basis for theory regarding the phenomenon rather than moving toward an existing theory [60]. Analysis, therefore, began by creating word clouds for each set of responses to look for emergent trends in the data that might assist with coding schemes. Word clouds have been found to be a useful tool in determining trends and coding schemes for qualitative data [61,62]. Categories were generated from the data, therefore, in a manner consistent with inductive analysis [60].

Using steps outlined by Hsieh and Shannon [60], the responses for both open-ended items were read, and any words or phrases that suggested a reaction related to the perception or efficacy of the manipulative were highlighted. The highlighted portions were then reviewed to identify common ideas among the highlighted portions. The common ideas were reviewed to identify, categorize, and label themes. Next, the responses were reviewed to bring related categories together and ensure there are categories for response data that do not fit the previously established categories. Lastly, the final list of categories was evaluated and arranged into a hierarchical structure based on how often responses occurred within the category.

## 4. Results

*4.1. Knowledge Acquisition and Retention*

A total of 112 treatment students had both pre- and post-test scores (see Table 1). The matched pair *t* tests indicated significant differences in all three Algebra 1 classes and in the Geometry A and C classes. Inspection of the worksheets from the Geometry B class determined that three students had multiplication errors on the procedural items, which caused their post-test scores to be lower than their pretest scores.

To examine the effect size for each class, Cohen's *d* was calculated. The effect size for the Geometry B class provided evidence of a small effect (0.202). The effect sizes for the Algebra A, Algebra B, Geometry A, and Geometry C classes provided evidence of a medium effect (ranging from 0.526 to 0.842). The Algebra C (inclusion) class provided evidence of a very large effect (1.378).

The results support that instruction using the CRA approach with a mathematical manipulative had a statistically significant increase in knowledge retention. The quantitative analysis provides evidence of significant knowledge gain after using the manipulative. These results support the hypothesis that knowledge transfer and retention can be facili-

tated by a manipulative that minimizes dual representation [57]. These results also support discussions that hands-on experiences assist students with LDs in their understanding of how numerical and abstract concepts operate at a concrete level [20,63,64].

**Table 1.** Matched pair *t* test and Cohen's d for the pretest and delayed post-test (N = 95).

| Class | *n* | *M* diff. | *SD* | *t* | *p* | Cohen's *d* |
|---|---|---|---|---|---|---|
| Algebra A | 22 | −1.272 | 1.9623 | −3.04 | 0.0031 | 0.701 |
| Algebra B | 22 | −0.9773 | 2.0206 | −2.27 | 0.017 | 0.526 |
| Algebra C (Inclusion) | 25 | −1.88 | 1.8044 | −5.21 | <0.0001 | 1.378 |
| Geometry A | 10 | −0.8 | 1.2517 | −2.02 | 0.037 | 0.842 |
| Geometry B | 18 | −0.5278 | 1.48 | −1.51 | 0.0743 | 0.202 |
| Geometry C | 15 | −0.8333 | 1.5079 | −2.14 | 0.025 | 0.744 |

*4.2. Perception of Manipulative Effectiveness*

The initial word cloud coding process indicated that students' responses focused on the value of the manipulative (e.g., helped, easier, understand, and better) and the attributes of the manipulative/activity (e.g., visual, way, and squares). Three simple categories were defined to examine how the students perceived the effectiveness of the lesson in developing an understanding of simplifying square roots: positive, neutral, and negative responses. The first category was identified as positive responses and included clear indications that the student found the manipulative helpful, enjoyable, easy, or simple were coded as having a positive response. The second category was identified as neutral responses and included indications that the student found the manipulative somewhat useful, that there was some confusion, or that the student needed more practice. The third category was identified as a negative response and included indications that the student did not find the manipulative useful or that they found the activity/method overly confusing.

Among all six treatment classes (*n* = 115), the majority of students (80%) made positive comments, whereas the remaining students were split among neutral (11.3%) and negative comments (8.7%). See Table 2 for complete data.

**Table 2.** Student perceptions of manipulative effectiveness (N = 115).

| Class | Positive | | Neutral | | Negative | |
|---|---|---|---|---|---|---|
| | *n* | % | *n* | % | *n* | % |
| Algebra A | 14 | 87.5 | 2 | 12.5 | 0 | 0 |
| Algebra B | 20 | 87 | 2 | 8.7 | 1 | 4.3 |
| Algebra C (Inclusion) | 9 | 36 | 8 | 32 | 8 | 32 |
| Geometry A | 13 | 100 | 0 | 0 | 0 | 0 |
| Geometry B | 17 | 94.4 | 1 | 5.6 | 0 | 0 |
| Geometry C | 19 | 95 | 0 | 0 | 1 | 5 |

Within the inclusion class, however, nine students (36%) had responses coded as positive, eight students (32%) had responses coded as neutral, and eight students (32%) had responses coded as negative. The responses coded as positive included phrases such as "visual understanding" and "hands-on", but there were fewer positive responses than the other responses. The majority (64%) of the students in the inclusion class were coded as having either neutral or negative perceptions of the lesson. Further analysis of the neutral and negative responses was conducted to find sub-themes. The predominant sub-themes from these responses were a lack of confidence and an aversion to being challenged. The responses coded as a lack of confidence showed a negative outlook on learning and mathematics. One student responded with "I don't remember nothing", while another responded with "Honestly it made me confused. I didn't really understand it, so show it

to the honors class, they'll understand". The responses coded as having an aversion to being challenged all referenced being confused or that the process was too long and had too many steps. One student responded, "No because it's too many steps", while another responded, "Not really. It takes time for me to know the steps and the process. Factors and square roots make it more confusing for me with all the steps".

The qualitative analysis provides some insight into students' perceptions of how this manipulative was used to deliver effective instruction. The Algebra I inclusion class showed mixed perceptions of the effectiveness of using a mathematical manipulative. The first level of coding found that the responses were almost evenly distributed among the three themes. The sub-themes provided insight into why the students had either a neutral or negative response to using the manipulatives. The manipulative challenged the students to see connections between the geometric representation and a numerical process for simplifying square roots (i.e., identifying perfect square factors). The researchers believe that the students were viewing the lesson in a direct instruction context, typical in mathematics instruction, where everything that is performed in the lesson is modeled by the teacher and then reproduced by the student, step by step. It seems that it was unclear to these students that using the manipulative was to gain a better understanding of simplifying square roots rather than to replicate the entire activity on their own.

When it comes to students' development of knowledge, the majority of the students stated that they were either unsure or did not believe that they would gain and retain knowledge from the lesson. On the contrary, the quantitative analysis provided evidence of significant gains. Perhaps the implementation of CRA, an unfamiliar instructional approach, impacted their perception of knowledge gain and retention. The lesson was designed so that students had to make sense of the process of simplifying square roots visually and progress to making connections to the numerical method while the teacher facilitated the activity. Many students perceived this process as confusing, with several stating that they needed repeated practice before they would retain the knowledge (e.g., "If we continue to practice, then yes! If not, no!").

A need for repeated practice is a hallmark of direct instruction. Based on questionnaire responses, students in the inclusion class seemed to believe that mathematics learning occurs through teacher-led instruction followed by repeated practice. Often, students perceive confusion during a direct instruction lesson to be associated with not learning [65,66]. Perhaps the extensive prior use of direct instruction with students with LDs made them less comfortable with the experience of using manipulatives to develop relational understanding. Providing these students with more information about the CRA instructional approach and how it compares to direct instruction at the beginning of the lesson may have avoided some of the confusion and misunderstood expectations of what students were learning. Implementing the CRA instructional approach regularly may also help students with LDs see the value of this approach. Further research is needed to test these hypotheses.

## 5. Discussion

Providing high school mathematics students with LDs with an effective instructional approach to develop connections between concrete representations and abstract procedures is within mathematics teachers' reach. Unlike studies presented by Maccini et al. [46], this study provides support that the CRA instructional approach, along with a manipulative that minimizes dual representation, can be effective in both knowledge acquisition and retention in a high school setting. It is important to highlight, however, that the majority of students expressed that they did not believe that this approach was helpful. The students in the larger study, in contrast, had predominately positive statements about the efficacy of the CRA instructional approach.

This study adds to the limited research on using manipulatives in secondary school mathematics classrooms and suggests that the CRA instructional approach needs to be explored further with other abstract topics. As several topics in the typical high school curriculum relate to the properties of squares, this manipulative has the potential to

further assist students with abstract thinking across mathematics courses. Two such topics are proportional reasoning related to the area and side length of squares and the converse of the Pythagorean theorem. Further expansion of manipulatives and the CRA approach in secondary school mathematics classrooms also aligns with successful practices demonstrated by Singapore.

This approach allows students to focus on the properties of the manipulative and support their progress to abstract thinking. As many abstract mathematical concepts are grounded in geometric representations, other similar manipulatives should be developed and explored. As further studies emerge that support using manipulatives for abstract thinking, the CRA instructional approach in high school classrooms should increase. In turn, all high school students will have multiple opportunities to develop their relational understanding by making clear connections between visual representations and abstract procedures.

There are several limitations to this study that should be highlighted and discussed. The first limitation was the lack of a control group for the inclusion class. Data from a control group could have compared the CRA approach to traditional instruction. Since the identification of the inclusion class happened after data analysis, data from a control group could not be collected. Including a control group in a future study could provide more clarity regarding student retention when using the CRA approach. Additionally, because the open-ended questionnaire was anonymous, we were unable to link specific students' perceptions of the manipulative lesson with their post-test scores to determine any patterns of student improvement and their perceptions. Future research should create a participant number for each student so these data can be linked.

This study was also limited in what could be determined related to the student's perception of mathematics and the learning of mathematics. The open-response questionnaire provided the researchers with insight into the students' perceptions of the instructional approach and the manipulative as they related to learning a mathematics topic, but not the students' perceptions of themselves as a learner of mathematics. The limited data did, however, provide glimpses into students' perceptions of the purpose of the activity. Future research may indicate the role of the instructor in metacognitive modeling (e.g., "think alouds" [67]) to support students' conceptual understanding of the mathematics content, in general, and the CRA activity, specifically. Creating a classroom culture in which students not only identify confusions but discuss them will also increase metacognition, which can sometimes be neglected when the focus is on a hands-on or interactive lesson [67]. Normalizing these confusions could influence students' willingness to try new activities or persevere when a lesson seems particularly challenging or unhelpful, as was the case with some participants.

Finally, follow-up questionnaires or semistructured interviews could provide further insight into high school inclusion students' self-efficacy as a learner of mathematics after partaking in lessons using the CRA approach, particularly as the students in the present study had substantial misperceptions of their learning with this lesson. Teachers and researchers could conduct clinical interviews, which would elicit information related to students' problem-solving processes that are difficult to obtain through other methods [68,69]. Collecting more qualitative data would allow for a more robust analysis and provide awareness of how different instructional approaches impact students' self-efficacy as learners of mathematics. This type of information could be helpful to teachers, researchers, and policymakers as they consider implementing different instructional approaches throughout the mathematics curriculum.

**Author Contributions:** Conceptualization, S.F.B.; methodology, S.F.B. and S.K.P.; validation, S.F.B. and S.K.P.; formal analysis, S.F.B.; investigation, S.F.B.; resources, S.F.B.; writing—original draft preparation, S.K.P.; writing—review and editing, S.K.P.; visualization, S.K.P.; supervision, S.F.B.; project administration, S.F.B. All authors have read and agreed to the published version of the manuscript.

**Funding:** This research received no external funding.



**Institutional Review Board Statement:** The study was conducted in accordance with the Declaration of Helsinki and approved by the Institutional Review Board of the University of South Carolina (protocol code Pro00043889 and 8 September 2015).

**Informed Consent Statement:** Informed consent was obtained from all subjects involved in the study.

**Data Availability Statement:** The data presented in this study are available on request from the corresponding author.

**Acknowledgments:** We would like to acknowledge and thank the teachers at Chapman High School in Spartanburg, South Carolina, for opening their classrooms and providing students with the opportunity to explore different ways to think about and learn mathematics.

**Conflicts of Interest:** The authors declare no conflict of interest.

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
