# Peer review of "Concrete–Representational–Abstract (CRA) Instructional Approach in an Algebra I Inclusion Class: Knowledge Retention Versus Students’ Perception"

_education, doi:10.3390/educsci13101061_

Round 1

Reviewer 1 Report

II believe that studies like this (simple but to the point and empirical) should be more published in the field of mathematics education research. 

There are a few typos and grammar errors. Please thoroughly review and edit accordingly. It would help if you have a third person to review and give you a feedback. 

Reviewer 2 Report

A very interesting article describing research that produced an interesting by-product. The article is written in a clear and understandable way and contains all the necessary information. I'm not a native speaker, but I feel that the language is sometimes less formal than is usual for scientific articles. I would consider it a good idea to add more analysis to the article, linking qualitative and quantitative analysis, i.e. the distribution of student improvement within the three categories determined qualitatively. I would also recommend adding to the analyses by interpreting the data graphically.

I appreciate that the authors are aware of the limitations of the data and appropriately suggest repeating the experiment.
